# OpenReview forum: "Manifold Learning for Adversarial Robustness: A Geometric  Defense Framework for Vision-Language Models"
_TMLR — Rejected by TMLR_

### Review · Reviewer_BhfJ · 2025-10-30

**Summary Of Contributions:**

Summary:
The paper formulates adversarial defense as a manifold learning problem where they use a Riemannian geometric framework to detect adversarial perturbations and correct the anomaly features using geodesic projection. This work provides an end-to-end protection and eliminates the need of adversarial training. Empirically, they verify that the proposed curvature-based detector achieves reliable detection across diverse tasks. The proposed method also shows superior robust accuracy under both vision, language and joint multimodal attack.

Strengths
1. The proposed method shows good performance in detecting adversarial examples and adversarial defense across different types of attacks.

Key Weaknesses
1. Weak Motivation and Unsubstantiated Theoretical Claims.
2. Ambiguous Training Procedure

**Audience:**

Yes

**Audience Explanation:**

The paper is mostly relevant to the audience working on adversarial robustness/training and safety of vision-language models.

**Claims And Evidence:**

No

**Claims Explanation:**

Weaknesses

1. Weak Motivation and Unsubstantiated Theoretical Claims.

The paper's core motivation for using a Riemannian geometric framework is not fully convincing. Section 3 posits that existing methods lack "principled mathematical foundations" and that adversarial perturbations manifest as "high-curvature anomalies"  on the learned manifold. However, the paper fails to provide a clear theoretical or intuitive justification for why this premise holds, especially for fused multimodal embeddings. It is not self-evident why input-space perturbations (like pixel changes or word substitutions) should consistently and measurably distort the geometry of the learned feature space in the specific way (i.e., high Ricci curvature) that the method relies on for detection.

Furthermore, the paper makes a strong claim about providing "provable geometric guarantees". The claim is not supported by the manuscript. The evaluation is entirely empirical, comparing robust accuracy on test sets. There are no theoretical bounds or formal proofs provided to substantiate the claim of provable robustness. This is a significant discrepancy between the paper's stated contributions and the evidence provided.

2. Ambiguous Training Procedure

The training methodology in Section 4.5 lacks critical details, making the results difficult to interpret. First, the method introduces several learnable components: the projection matrix $W$ and the neural networks $\phi_L$ and $\phi_w$. The paper does not explicitly state whether these are all trained jointly from scratch using the unified loss $\mathcal{L}_{total}$, or if some are pre-trained or frozen.

Moreover, the paper claims that the method requires zero attack examples during training. However, the geodesic consistency loss, $\mathcal{L}_{geo}$, is defined as the Riemannian distance between the original features ${f_m^i}$ and their "corrected" geodesic projections $\hat x_i$. The paper states this loss penalizes "large corrections needed for adversarial inputs". This implies that adversarial examples must be fed into the model during training to generate a meaningful, non-zero $\mathcal{L}\_{geo}$ loss. If the model is trained only on clean data, the original features $f_m^i$ would presumably already lie on the "safe" manifold, resulting in a negligible distance $d_G(f_m^i, \hat{x}_i)$ and rendering this loss term ineffective. The authors must clarify:

- Is the model trained exclusively on clean data?
- If yes, how does $\mathcal{L}_{geo}$ learn to penalize unseen adversarial deviations?
- If no, and adversarial examples are used to compute $\mathcal{L}_{geo}$, then the paper's central claim of not using adversarial examples is incorrect and must be revised.


3. Other issues
- Computation of Ricci curvature: Can the authors provide more details in computing the local Ricci curvature in Eq. (10)?
- Inconsistency in Evaluated Models: The experimental setup in Table 2 introduces three models: CLIP, BLIP-2, and LLaVA-7B. However, all quantitative results presented in the main tables (Table 4 and Table 5 ) and figures (Figure 2-6) appear to be exclusively for the CLIP model. The authors must either provide these results or remove the other models from Table 2 and temper their claims.
- Missing and Ambiguous Citations:
   - Mathematical Foundations: Section 4 introduces several concepts from Riemannian geometry (metric learning, geodesic projection, Ricci curvature) . To allow a proper evaluation of the paper's novel technical contributions, the authors should provide citations for standard, pre-existing geometric techniques. This would help distinguish the paper's novel components from the application of established methods.
   - Baselines and Attacks: The paper provides no citations for the defense baselines (e.g., AdaShield, ECSO, SafeEraser) or the attack methods (e.g., TextBugger, BAE, PGD, AutoAttack , and the joint attacks ) used in the evaluation. This is a significant omission that prevents proper contextualization and verification of the experimental results.
- Notational and Metric Ambiguity:
   - Dimensions: In Section 4.4, the dimensions of the computed values are unclear. The paper must specify the shape of the Ricci curvature $\kappa(\hat{x})$ 9 (is it a scalar per feature vector?) and the final output $f_{out}$.
  - Equation (12): The meaning of the index $j$ in the attention equation $f_{out}^{i}=\sum_{j}\alpha_{ij}\hat{x}_{j}$ is ambiguous. It is unclear if $j$ iterates over the $d$ feature dimensions or represents the $j$-th example in the data. This is critical for understanding the mechanism.
  - Curvature Ratio: This metric is reported with two drastically different scales. In Table 4, it has values like 1.02, 3.46, and 4.91. In Figure 4, it is reported as a percentage score of 78.4%. The paper must provide a clear, mathematical definition for "Curvature Ratio" and explain how these different values are computed and related.

**Requested Changes:**

Please see the weaknesses above.

---

### Review · Reviewer_Fxtv · 2025-11-02

**Summary Of Contributions:**

This paper provides a geometric framework to defense advasarial attack in vision language models. The main innovation lies in shifting the focus from attack-pattern memorization and adversarial training to learning the geometric manifold structure of clean multimodal embeddings. The method introduces a Riemannian manifold–based approach consisting of four components:

1. Cross-modal feature fusion to obtain multimodal embeddings.
2. Riemannian metric learning to characterize the geometry of representations.
3. Geodesic projection to correct perturbed features by projecting them back onto the manifold.
4. Curvature-based attention to suppress adversarial regions.

The proposed methodology and underlying intuition are intriguing. However, the paper lacks critical details and clarity, making it challenging to fully evaluate the method’s feasibility and reproducibility. Some major issues are listed as follow:

1. The framework is not clearly described or illustrated. There is no overview figure to visually summarize the proposed method, which makes it difficult to grasp the overall approach at a glance. Additionally, while the authors introduce several learnable components, including neural networks and parameters, they do not provide explicit details regarding their architectures or configurations. For example, neural network $\phi_{L}, \phi_{w}$, and learnable basis vector $v_k$. Are $\{v_k\}$ learned from input features $x$ or they are learnable parameters? If the former, what is the network used to learn them?

2. A lot undefined and wrongly defined notations. Equation (1) should be $f_lW$ based on the shape. Equation (4), $\tilde{L}$ is not defined. Based on my understanding, it should be $\tilde{L} = \phi_L(f_m)$. But then it contradicts with the sentence "where $\phi_L$ is a neural network producing $L$ with positive diagonal". Section 4.3, the authors use another notation $x$ for inout features, which I assume should be the same as $f_m$ with a potential to be adversarial. However, the dimension of $x$ should be $R^{N_v\times d}$ instead of $R^d$. This duplicated notations leads to confusion in equation (14), where it's better to use $x$ instead of $f_m$ for features before projection.

3. No intuition or theoretical guarantee is provided. The paper lacks a theoretical guarantee that the proposed framework can effectively achieve its intended goals. Furthermore, the authors do not offer a clear intuitive explanation of why or how their approach should work. As I understand it, the method is based on the hypothesis that visual and textual features from adversarial samples are not well aligned. However, this hypothesis is neither substantiated nor is it demonstrated that the proposed framework can reliably measure or address such misalignment.

4. Experiment design. The authors claim the proposed method efficiently handles all three models: CLIP (ViT-B/16) , BLIP-2 utilizes with DeepSpeed ZeRO-2, and LLaVA-7B. However, the experimental results are presented only for CLIP (ViT-B/16). Additionally, for the CLIP baseline, it is unclear whether the authors used a pretrained CLIP model or trained it from scratch. The baseline performance reported is noticeably lower than that in [1], and even below the few-shot results reported in [2]. Furthermore, [1] has reported that ViT based CLIP performs worse than ResNet based CLIP. Is there a specific reason why the authors choose ViT base?

5. Relevant works not discussed. The authors do not discuss the broader literature on manifold adversarial robustness. This area features substantial prior work and foundational methodologies, like [3,4,5,6].

[1] Sheng Shen, Liunian Harold Li, Hao Tan, Mohit Bansal, Anna Rohrbach, Kai-Wei Chang, Zhewei Yao, and Kurt Keutzer. How much can clip benefit vision-and-language tasks? In ICLR 2022.

[2] Song, Haoyu, et al. "Clip models are few-shot learners: Empirical studies on vqa and visual entailment." arXiv preprint arXiv:2203.07190 (2022).

[3] Wenjia Zhang, Yikai Zhang, Xiaoling Hu, Mayank Goswami, Chao Chen, Dimitris Metaxas: "A Manifold View of Adversarial Risk", in International Conference on Artificial Intelligence and Statistics (AISTATS), 2022

[4] Wenjia Zhang, Yikai Zhang, Xiaoling Hu, Yi Yao, Mayank Goswami, Chao Chen, Dimitris Metaxas:"Manifold-driven decomposition for adversarial robustness", in Frontiers in Computer Science, 2024

[5] Lin WA, Lau CP, Levine A, Chellappa R, Feizi S. Dual manifold adversarial robustness: Defense against lp and non-lp adversarial attacks. Advances in Neural Information Processing Systems. 2020;33:3487-98.

[6] Xiao J, Yang L, Fan Y, Wang J, Luo ZQ. Understanding adversarial robustness against on-manifold adversarial examples. Pattern Recognition. 2025 Mar 1;159:111071.

**Audience:**

Yes

**Audience Explanation:**

Despite some issues with clarity and completeness, the paper’s focus on manifold learning for adversarial defense addresses an area of substantial interest to the TMLR audience.

**Claims And Evidence:**

No

**Claims Explanation:**

As stated in the weaknesses, the paper lacks both clarity and correctness.

**Requested Changes:**

1. Please address the issues with notation and framework clarity as outlined in the weaknesses.

2. Provide theoretical justification, illustrative examples, or at least a clear explanation of why the proposed method is expected to work.

3. Include experimental results for the additional vision-language models mentioned, or adjust your claims accordingly. Additionally, improve the selection and reporting of baselines for fair comparison.

4. Discuss the broader literature on manifold adversarial robustness.

---

### Review · Reviewer_CSYC · 2025-11-25

**Summary Of Contributions:**

This paper studies the defense for multimodal large language models (MLLMs). It claims that current methods rely on expensive adversarial training that requires attack optimization and also lacks principled mathematical foundations for characterizing adversarial robustness. Main contributions include (1) presenting a Riemannian geometric defense framework for MLLMs. Specifically, the proposed framework learns metric tensors to recognize clean feature geometry and detects adversarial perturbations via Ricci curvature analysis. (2) The framework eliminates the requirement of adversarial training by learning geometric invariants rather than memorizing attack patterns. (3) Experimental results show that the proposed framework achieves superior robust accuracy on VQA v2.0 across multiple types of attacks.

Strengths:
- The application of Riemannian geometry to MLLM defense is a novel application.
- The method does not require expensive adversarial examples during training, which would greatly reduce training cost.
- The paper reports substantially improvements over baselines.

Weaknesses:
-  While training is efficient, the inference-time calculation of curvature and metric tensors  that might be prohibitive for real-time applications.
- The specific implementation of the curvature calculation (Eq. 10) is somewhat abstract in the text. Calculating the full Ricci tensor and its trace usually involves computing Christoffel symbols and derivatives of the metric, which is expensive. The paper needs to clarify if approximations are used.
- Unclear Method Description: The methodology section lacks sufficient detail for understanding. For example, The paper does not explain how Eq. 9 approximates a true Riemannian geodesic, nor does it justify why this linear formulation holds on a curved manifold. It is unclear exactly where in the model architecture this defense is inserted (e.g., only at the final fusion layer? At every transformer block?).

**Audience:**

Yes

**Audience Explanation:**

The topic of adversarial robustness for Multimodal LLMs is timely and important. If the methodological details are clarified and the claims properly scoped, the geometric perspective offers an interesting direction for the community.

**Broader Impact Concerns:**

No specific ethical concerns.

**Claims And Evidence:**

No

**Claims Explanation:**

- Training Efficiency: The authors claim the method avoids "expensive adversarial training" (AT). However, there are no experimental results comparing the training time/cost (wall-clock time or FLOPs) of GeoDefend vs. AT. Given that GeoDefend requires computing metric tensors, Cholesky factorizations (Eq. 4), and geodesic consistencies (Eq. 14) at every training step, it is not obvious that it is computationally cheaper than standard AT. Evidence is missing to support the claim that this is an efficient alternative.
- "Guarantees" and "Certified Defense": The paper uses strong language regarding theoretical backing, such as "certified defense framework" (end of Section 2) and "geometric guarantees" (Section 4). However, the paper provides no formal proofs, certified radius bounds, or theoretical guarantees that the defense holds for all perturbations within an ϵ-ball. The "verification" provided is purely empirical (on specific attack benchmarks), which contradicts the claim of a "certified" or "guaranteed" defense.
- The paper claims to learn the "natural manifold structure," but offers no analysis (e.g., intrinsic dimension estimation, visualization of the learned metric spectrum) to prove that the learned tensor G actually corresponds to the semantic manifold of the data rather than just overfitting to the training distribution.

**Requested Changes:**

Please refer to the above sections.

Additional strengthening changes:
- Show the sensitivity of performance to the geodesic basis dimension K.

---

### Decision · Action_Editor_P3Ru · 2026-01-27

**Recommendation:** Reject

**Audience:**

No

**Audience Explanation:**

Due to limited empirical evidence, the message of the paper is not trustworthy.

**Claims And Evidence:**

No

**Claims Explanation:**

The paper proposes a Riemannian geometry-based defense for MLLMs but cannot be accepted due to critical flaws identified by reviewers. These include insufficient implementation details, potentially prohibitive computational costs, and inadequate evaluation (limited to CLIP model only). Furthermore, while the method claims theoretical superiority, it lacks the actual theoretical guarantees to support this. No rebuttal was submitted to address these issues.